# Does Environmental Regulation Promote the Infrastructure Investment Efficiency? Analysis Based on the Spatial Effects

**DOI:** 10.3390/ijerph20042960

**Published:** 2023-02-08

**Authors:** Maohui Ren, Tao Zhou, Di Wang, Chenxi Wang

**Affiliations:** 1School of Management Science and Real Estate, Chongqing University, Chongqing 400044, China; 2Research Center for Construction Economy and Management, Chongqing University, Chongqing 400044, China; 3School of Geographical Sciences, Southwest University, Chongqing 400715, China

**Keywords:** environmental regulation, infrastructure investment efficiency, Super-SBM model, spatial effects

## Abstract

Infrastructure investment plays a key role in steady economic growth. Investment in infrastructure is gradually increasing; however, large-scale infrastructure investment is also associated with efficiency problems and environmental problems, which need to be carefully examined. The entropy weight method and the Super-SBM model are implemented to measure environmental regulation and infrastructure investment efficiency, respectively; and the spatial Durbin model is applied to analyze the influence mechanism and spatial effects of environmental regulation on infrastructure investment efficiency. The results show that spatial agglomeration characteristics exist in both environmental regulation and infrastructure investment efficiency. Besides, environmental regulation can promote infrastructure investment efficiency in general, but it develops an inverted U-shaped relationship as the intensity increases. Lastly, the spillover effects of environmental regulation on infrastructure investment efficiency present a U-shaped relationship. The conclusion is that both environmental regulation and infrastructure investment efficiency in China increased from 2008 to 2020. Besides, moderate environmental regulation is beneficial to infrastructure investment efficiency and restrains spatial spillover, but strict environmental regulation appears to be the opposite. This research expands the literature on environmental regulation and production efficiency and provides a reference basis for formulating effective policies to improve infrastructure investment efficiency from the perspective of the ecological environment.

## 1. Introduction

Faced with the worsening global ecological environment, governments around the world have reached multilateral agreements [1], such as the United Nations Framework Convention on Climate Change (UNFCCC), Paris Climate Agreement, and the Sustainable Development Goals (SDGs). China has also suffered from several environmental issues during its economic development [2]. Air, soil, and water pollution have affected the quality of Chinese economic development and sustainable development [3]. In the global Environmental Performance Index (EPI) released in 2021, China ranked in the bottom 1/3 of the countries, and its air quality ranked in the bottom 1/4. Therefore, the Chinese government has issued a series of regulations to protect the environment, including the Clean Production Promotion Law (2013), Air Pollution Prevention Act (revised in 2016), Water Pollution Prevention and Control Law (2017), the Energy Conservation Law (2018), and Regulation on the Administration of Permitting of Pollutant Discharges (2021). These environmental policies have played an important role in various production fields in China, and the aim of this research is to investigate the role of environmental regulation in the field of infrastructure investment.

Infrastructure investment is the forerunner of economic development. In China, infrastructure investment is a key economic stimulus formalized in regular National Five-Year Plans [4]. According to the World Bank, by 2040, China’s investment in infrastructure will reach RMB 28 trillion, accounting for 30% of the global total [5]. However, although urban economic development is significantly related to infrastructure construction [6], there is an upper limit to the economic drive from large-scale infrastructure construction [7]. Large-scale infrastructure construction is often associated with a large amount of gravel, sand, wood, and energy consumption, which produces environmental hazards such as wastewater and exhaust gases [8]. Besides, the built infrastructure also has a negative impact on the environment; for example, the energy used over the life cycle of a 1 km road section with two lanes is 11.38 TJ [9]. Some researchers have focused on exploring the performance of infrastructure investment [4,10,11], finding that China’s infrastructure investment efficiency is not compatible with the quality of urbanization [12].

In addition to the scale and performance of infrastructure investment, scholars also focus on the spatial distribution and spatial evolution of infrastructure investment efficiency. Regional inequality in China’s municipal public infrastructure investment efficiency gradually exacerbates [12]. Eighty-six percent of all Chinese cities possess low infrastructure utilization efficiency, and only eleven percent of cities exhibit continuous satisfactory performance [5]. Studies have also been carried out worldwide, which found that the infrastructure investment efficiencies of Central European countries, New Zealand, and Japan are the highest while those of Eastern European countries, Russia, Turkey, and Mexico are the lowest [10]. Road infrastructure productivity has a significant spatial spillover effect [13], and labor productivity in the agricultural sector exhibits spatial heterogeneity [14]. The DEA method is widely used when calculating infrastructure investment efficiency [4,5,12].

Existing literature focuses not only on infrastructure investment but also on the environmental issues associated with it and the current environmental regulations. Some studies have explored the environmental and efficiency issues related to infrastructure construction. The construction of power, industrial, and transportation infrastructure leads to air pollution and endangers the health of residents in most cities [15]. Unreasonable infrastructure policies can lead to the wastage of resources, hinder urban development, and even cause complex social problems [11,16]. Environmental pollution from infrastructure construction and operation with negative externalities is the direct object of environmental regulation [17]. At present, research on environmental regulation is developing gradually. Strict environmental regulation may stimulate green innovation [18,19]. The relationship between environmental regulation and technological innovation efficiency is U-shaped [20] or inverted U-shaped [21], and environmental regulation has a significant spatial spillover effect on green innovation efficiency [22]. Weak and medium environmental regulation promotes energy consumption structure optimization, while strong environmental regulation inhibits this optimization [23].

The above literature has indicated that there is a correlation between environmental regulation and infrastructure investment, which may exert impacts on infrastructure investment efficiency. It is worth noting that the influence of environmental regulation on efficiency has been studied extensively. Environmental regulation could promote production efficiency [24,25,26] or inhibit it [2,27,28,29]. Some studies have shown that the relationship between the two may not be linear [20,30,31,32], or may not exist at all [33]. Environmental regulation can affect production efficiency by changing the inflection point and the curvature of the EKC [3,34]. Strict environmental regulation can push the inflection point of the EKC to appear earlier by promoting technological progress and industrial transfer [35], which is strongly related to production efficiency [36]. Furthermore, from the spatial perspective, environmental regulation affects efficiency by influencing the locations of enterprises [37,38]. In previous studies, the compliance cost hypothesis held that environmental regulation would reduce production efficiency by directly increasing the cost of enterprises [27,28,39], but the Porter hypothesis proposed that strict environmental regulation would promote production efficiency by stimulating technological innovation [24,26,40].

Based on the above literature review, we found some shortcomings in the existing research. Most current research has analyzed the temporal and spatial characteristics of infrastructure investment efficiency, without considering the sustainability of infrastructure construction. Some studies investigated the relationship between environmental regulation and efficiency from the perspective of space but ignored the spatial spillover effects and spatial interactions. Few studies have focused on the efficiency and environmental issues of large-scale infrastructure investment, and evidence from empirical data is lacking. Therefore, research on the relationship between environmental regulation and infrastructure investment efficiency from the spatial perspective is essential, and a theoretical discussion on the influence mechanism is imperative. This study aims to empirically examine the relationship between environmental regulation and infrastructure investment efficiency in China from 2008 to 2020 and to explore the spatial interaction mechanism. Accordingly, the spatial econometric method is employed to analyze the spatial spillover effects, and the Super-SBM model—an improved form of the DEA model—is established to measure the infrastructure investment efficiency. 

Our research mainly provides contributions in the following three aspects: Firstly, make further and deeper research on environmental regulation. Existing literature focuses mainly on the characterization and empirical analysis of environmental regulation, and most of them implement a single index to measure environmental regulation. On this basis, we build a sound measurement system of environmental regulations and empirically investigate the spatial and temporal distribution of environmental regulations. Secondly, it provides a more in-depth study of the spatial and temporal distribution and spatial effects of production efficiency. The existing research mainly focuses on the measurement and spatial distribution of production efficiency, but the constructed index system is deficient and does not consider the spatial spillover effect. In this study, a sound index system is established to measure a specific production efficiency (infrastructure investment efficiency), and its spatial spillover effect is considered and decomposed. Finally, this study enriches the research on the influence mechanism of environmental regulation on production efficiency. Existing studies mainly measure the linear impact of environmental regulations on a certain factor, without considering the spatial effects of environmental regulations. On this basis, we also consider the nonlinear effects and spatial effects of environmental regulations, which expands the research on relevant theories.

The significance of this study is as follows. Firstly, it expands the literature on environmental regulation, production efficiency, and their spatial interaction and enriches the classical theories. Secondly, establishes sound index systems to measure the infrastructure investment efficiency and environmental regulation, which provides a tool for the government to measure the intensity of the environmental policies and the utilization of infrastructure investment. Finally, the empirical results and conclusions provide a reference basis for formulating effective policies to improve infrastructure investment efficiency from the perspective of the ecological environment.

The rest of this article is organized as follows. In Section 2, a theoretical discussion on the influencing mechanisms between environmental regulation and infrastructure investment efficiency has been conducted. In Section 3, quantitative analysis methods, the empirical regression model, and data sources are described. Section 4 and Section 5 cover the analysis and discussion of the results, respectively. Finally, conclusions are drawn in Section 6.

## 2. Theoretical Framework and Research Hypothesis

Through a literature review, it is found that environmental regulation may not only be associated with infrastructure investment efficiency but also interact with it in space. The construction and operation of infrastructure are the direct targets of environmental regulation. In addition, environmental regulation will affect the resources needed for infrastructure construction and operation.

### 2.1. Direct Influence

Infrastructure construction and operation consume a lot of energy [9]; therefore, environmental regulation will directly guide enterprises in terms of reducing energy consumption. Besides, the government will offer green economy subsidies to companies that adopt cleaner production technologies [41]. For example, tax deductions for the purchase of cleaner production equipment. However, businesses in the infrastructure industry must purchase cleaning equipment, repair environmental damage, and improve the production process to comply with environmental regulations [42]. Therefore, when the saved costs in the above cases are more than the compliance costs, environmental regulation plays a facilitating role; otherwise, it plays a negative role. This phenomenon may lead to nonlinear effects of environmental regulation.

### 2.2. Indirect Influence

#### 2.2.1. Green Technology Innovation

When strong environmental regulation is implemented, enterprises will increase their investment to motivate green technology innovation and reform management models [30], the results of which indirectly improve the production and operation efficiency of enterprises.

#### 2.2.2. Industrial Structure

With the strengthening of environmental regulation, some enterprises, with high pollution and emissions, are eliminated due to the rising green cost and policy requirements. As a result, the leading enterprises, with higher production technology, gradually dominate the market, which is conducive to the improvement of production efficiency. Furthermore, enterprises tend to concentrate production in areas with eased environmental regulation, and the agglomeration of polluting enterprises is conducive to the formation of a synergistic effect on pollution control [18].

#### 2.2.3. Opening Up

Reasonable opening-up policies and environmental policies are conducive to increasing R&D investment, enhancing regional comparative advantages, and stimulating foreign investment [43]. Direct foreign investment guarantees R&D investment in enterprises. However, strict environmental regulation will form investment barriers; international companies would prefer to invest in highly polluting projects in regions with relatively loose regulations, which will lead to a wastage of resources and reduce infrastructure investment efficiency [20]. 

### 2.3. Spatial Effects

#### 2.3.1. Spatial Signal Transmission Mechanism

Regional differences exist in the spatial distribution of environmental regulation [44], and this research proposes the mechanism of environmental signal transmission, which leads to spatial heterogeneity in the impact of environmental regulation on infrastructure investment efficiency. At first, the environmental regulation intensity is low, and the pollution is strong in a certain region; however, with pollution spillover, environmental conditions in this region and neighboring regions will deteriorate, leading to alarm in neighboring regions. As a result, neighboring regions will increase the intensity of environmental regulation. However, as the geographical location becomes further away, the spillover of pollution will gradually decrease, and the pollution alert signals will decay. Finally, distant regions will believe that low-intensity environmental regulation will not lead to serious pollution, and then select low-intensity environmental regulation strategies, leading to the spatial effects of environmental regulation on the efficiency of infrastructure investment (see Figure 1). 

#### 2.3.2. Spatial Spillover Effects

Changes in the environmental regulation intensity in a certain region produce spatial compensation effects or spatial substitution effects on infrastructure investment efficiency in adjacent areas. Compensation effects refer to the fact that regions with stricter environmental regulation take the lead in terms of green innovation, such that regions with weaker environmental regulation must increase technological innovation inputs to maintain market competitiveness. Substitution effects refer to the fact that enterprises in regions with weaker environmental regulation directly introduce new technologies developed by enterprises in regions with strong environmental regulation to enhance their market competitiveness. When the compensation effects are stronger than the substitution effects, environmental regulation promotes positive spatial spillover effects on infrastructure investment efficiency. The opposite is true when the compensation effects are weaker than the substitution effects. 

In summary, the influence mechanism is shown in Figure 2.

### 2.4. Research Hypothesis

Based on the above theoretical analysis, we propose the following research hypotheses:

**Hypothesis** **1.**
*Environmental regulation generally promotes the infrastructure investment efficiency.*


**Hypothesis** **2.**
*Nonlinear effects exist in the influence of environmental regulation on the efficiency of infrastructure investment, and too strong environmental regulation will inhibit infrastructure investment efficiency.*


**Hypothesis** **3.**
*Spatial agglomeration characteristics exist in both environmental regulation and infrastructure investment efficiency.*


**Hypothesis** **4.**
*Spatial effects also exist in the impact of environmental regulation on infrastructure investment efficiency.*


## 3. Data and Methods

### 3.1. Data

According to the availability of data, this study uses panel data from 30 provinces and municipalities in China from 2008 to 2020, excluding the data from Hong Kong, Taiwan, Macao, and the Tibet Autonomous Region. The data come from the China Environmental Statistics Yearbook (2009–2021), China Labor Statistics Yearbook (2009–2021), China Statistical Yearbook (2009–2021), and China Environmental Yearbook (2009–2021) each year. In addition, we refer to the databases of CNQI, China Economic Network (2009–2021), and EPS DATA. To ensure the stability of panel data, the average growth rate is adopted to complete the missing data from some provinces.

### 3.2. Measurement of Infrastructure Investment Efficiency

#### 3.2.1. Super-SBM Model

Existing efficiency measurement methods primarily include the stochastic frontier approach (SFA) and data envelopment analysis (DEA) [45]. SFA, a type of parametric method, implements the stochastic frontier production function to estimate efficiency. However, infrastructure investment is a complex system, and the production function of the input and output of infrastructure investment efficiency is unclear and variable. DEA is a nonparametric method that does not need to predefine the functional relationship between inputs and outputs. Therefore, DEA is widely adopted to measure infrastructure investment efficiency [4,5,10,12]. Charnes, et al. [46] first proposed the DEA model, including the traditional CCR model and BCC model. These models neglect the “relaxation” effect of factors; therefore, the efficiency measurement could produce errors. The Slacks-based measure (SBM) model proposed by Tone [47] effectively mitigates this issue; however, its calculation process contains multiple effective units. Tone [48] introduced the Super-SBM model on the basis of the SBM model to make up for the limitation of the SBM model in that it cannot distinguish effective decision-making units. The model definition is shown in Formula (1), where x and y represent input and output variables, respectively; m and s are the number of input and output indicators, respectively; $s_{i}^{-}$ and $s_{r}^{+}$ represent the relaxation variables of the input and output, respectively; and $\lambda_{j\;}$ represents the weight vector.
(1)$$\left(D_{\varepsilon }\right)\{\begin{array}{c} min\theta =\frac{1+\frac{1}{m}\sum_{i=1}^{m}\frac{s_{i}^{-}}{x_{ik}}}{1-\frac{1}{s}\sum_{i=1}^{m}\frac{s_{r}^{+}}{y_{rk}}}\\ s.t.\underset{j=1,j\neq k}{\sum }x_{ij}\lambda_{j}-s_{i}^{-}\leq x_{ik}\left(i=1,2,\ldots ,m\right)\\ \underset{j=1,j\neq k}{\sum }y_{rj}\lambda_{j}+s_{r}^{+}\geq y_{k}\left(r=1,2,\ldots ,s\right)\\ \lambda_{j}\geq 0,j=1,2,\ldots ,n\left(j\neq k\right),s_{i}^{-}\geq 0,s_{r}^{+}\geq 0 \end{array}$$

The Super-SBM model is implemented to measure infrastructure investment efficiency. The model follows the basic principles of the DEA model. When the research object contains multiple input and output factors, in order to measure whether the output corresponding to each input is efficient, it is necessary to build some kind of production-effective space that can reflect the optimal interaction of cost and income mapping. Then, the deviation range between the decision-making unit and the envelope surface is analyzed to measure the production efficiency. In the application of the model, emphasis is placed on the selection of input and output indicators, which will be explained in detail next.

#### 3.2.2. Index System

Infrastructure-related industries need to be identified before selecting indicators to calculate infrastructure investment efficiency. According to the World Development Report 1994, infrastructure includes permanent constructions, facilities, and equipment that provide services for economic production and benefit residents, including (1) public services, such as water supplies, sanitation, piped gas, sewerage, and electricity; (2) public works, such as dams and irrigation systems; (3) transportation projects, such as roads, airports, railways, urban transportation, and seaports; and (4) social service facilities, such as education, culture, sports, and health [49]. Further, the Industry Classification of National Economy (GB/T 4754-2017) adopted in China and the annual China Statistical Yearbook accurately identified infrastructure-related industries, such as water conservation, the environment, and public facility management industries. In previous literature, indicators for corresponding industries were also selected to calculate infrastructure investment efficiency [4,5,7,12]. Based on the three aforementioned points, this study selects some industries that fall under the category of infrastructure investment, as shown in Table 1 (scope of infrastructure investment). The overall structure of the infrastructure investment efficiency measurement index system is referred to by Liu, Wang, Zhang, Li, Zhao, and Li [12]. Infrastructure construction requires an investment of capital and labor, so the fixed investments of these industries are selected as the capital input indicator [12], and the corresponding numbers of employees of these industries are selected as the human capital input indicator [5,50,51]. 

In terms of output indicators, infrastructure investment can not only lead to the production of buildings but also drive economic development. The total GDP and average disposable income are selected to reflect the impact of infrastructure investment on economic development and residents’ lives [4,52,53]. In addition, we extend the evaluation system built by Liu, Wang, Zhang, Li, Zhao, and Li [12]. And seven sub-systems are selected to establish the infrastructure evaluation index system [4,6,7,10]. The time series global principal component analysis is used in SPSS 23 to calculate the urban infrastructure composite index, which is used as the third indicator to assess the specific effects of infrastructure investment, as shown in Table 1 (infrastructure evaluation index system). The index system to measure infrastructure investment efficiency is finally constructed, as shown in Table 2. To eliminate the influence of prices, fixed asset investment data are converted using the fixed asset investment price index. 

### 3.3. Measurement of Environmental Regulation

#### 3.3.1. Entropy Evaluation Method

Environmental regulation is an abstract concept describing the various environmental control measures in a region. However, the sources of pollution and corresponding control methods in each region differ, so there is no unified direct index to assess the concept of “environmental regulation”. Previous studies have adopted a number of different index weighting methods [54], such as the TOPSIS evaluation method, AHP evaluation method, and fuzzy comprehensive evaluation method. The entropy weight method, derived from the concept of entropy, is able to completely obtain the data itself without considering subjective factors. The relevant indicators of environmental regulation are objective statistical data; therefore, the entropy weight method is adopted to measure the intensity of environmental regulation [12,18,54,55,56].

“Entropy” is a physical description of the motion of molecules when heated, which can reflect the complexity of the data in the index. The index with more complex data contains more information, so the index value will appear more different and heterogeneous, and the internal effective information will be richer; accordingly, the index will be given a larger weight. The calculation rules for the entropy and the weight of an index are as follows.

(1) Data standardization: positive indicator, $x_{ij}{}^{\prime }=\left(x_{ij}-x_{min}\right)/\left(x_{max}-x_{min}\right)$; negative indicator, $x_{ij}{}^{\prime }=\left(x_{max}-x_{ij}\right)/\left(x_{max}-x_{min}\right)$. Here, $x_{ij}$ is the j-th indicator in the i-th year, $x_{max}$ is the maximum value of all indicators, and $x_{min}$ is the minimum value of all data of the indicator.

Since logarithms are used in the entropy method, normalized values cannot be used in the calculation directly. To reasonably solve the impact of negative numbers, the normalized value is translated as follows: $Z_{ij}=x_{ij}^{'}+A$, where $Z_{ij}$ is the translated value and A is the translated magnitude.

(2) Calculate the proportion of the j-th indicator in the index ($p_{ij}$): $p_{ij}=Z_{ij}/\overset{n}{\underset{i=1}{\sum }}Z_{ij}$, where n is the number of objects participating in the evaluation.

(3) Calculate the information entropy value ($e_{j}$):$\;e_{j}=-k\overset{n}{\underset{i=1}{\sum }}p_{ij}ln\left(p_{ij}\right)$*,*$\;k=\frac{1}{\ln\left(n\right)},\;e_{j}\geq 0$.

(4) Calculate the difference coefficient ($g_{j}$): $g_{j}=1-e_{j}$.

(5) Calculate the weight ($w_{j}$):$\;w_{j}=g_{j}/\overset{m}{\underset{j=1}{\sum }}g_{j}$ (j=1,2…,m), where m is the number of indicators.

(6) Calculate the environmental regulation intensity $F_{i}$*:*$F_{i}=\overset{m}{\underset{j=1}{\sum }}w_{j}p_{ij}$.

(7) The unit of $F_{i}$ is set as 10^−3^ to facilitate display and calculation.

#### 3.3.2. Index System

Most existing studies adopt surrogate indicators to measure the intensity of environmental regulation. Some scholars choose a single indicator to represent this variable, such as the ratio between pollution charges and the gross product value [57], regional environmental pollution control investment [51], or the amount of industrial pollution-elimination investment in the GRDP [35]. However, environmental regulation is a comprehensive concept, and some researchers have established an index system to measure it [55]. 

To comprehensively assess the current situation scenario of environmental regulation, this study constructs the following index system from four aspects: government administration, regulation effects, economic input, and residents’ green life quality. Levying pollution charges is an important means for the government to manage the environment [58]; therefore, the indicator enterprises pollution charges/GDP is used to measure the effectiveness of government administration. A number of studies have shown that environmental regulation is directly related to local economic development [2,51,59,60]; therefore, the indicator per capita GDP is adopted. Furthermore, economic investment in environmental pollution is represented by investment in environmental pollution control/GDP, which is also directly related to the level of regional economic development [51]. Moreover, environmental regulation can limit pollutant emissions and enhance the treatment of pollutants [57], the effect of which is reflected by the indicators of pollutant discharge amount (sulfur dioxide, solid waste, and sewage)/industrial added value, and the pollutant (sewage and garbage) treatment rate. Finally, environmental regulation is related to the ecological efficiency of urbanization and directly affects residents’ green life quality [50,52]. The indicators of per capita park green area and coverage rate of sanitary latrine, formulated into many provinces’ five-year plans, are suitable to measure the green life quality. The index system is shown in Table 3.

### 3.4. Spatial Econometric Method

#### 3.4.1. Spatial Correlation Test

The spatial correlation test is mainly implemented to test whether there exist spatial effects in both environmental regulation and infrastructure investment efficiency (Hypothesis 3). Spatial effects refer to the characteristics of the spatial agglomeration of variables, which provide conditions for the implementation of spatial econometric models [61,62]. The global Moran’s *I* test is implemented to reflect the overall spatial distribution characteristics of the research objectives. Global spatial autocorrelation diagnosis is evaluated by calculating Moran’s *I* and *p*-value. The value of Moran’s *I* is between [−1, 1]. If Moran’s *I* of the variable is greater than 0 and significant, it indicates that there exists a spatial aggregation feature; otherwise, it indicates that there is no spatial aggregation feature [63]. The global Moran’s *I* is calculated using the following formula:(2)$$I=\frac{n}{s_{0}}\frac{\sum_{i=1}^{n}\sum_{j=1}^{n}w_{i,j}z_{i}z_{j}}{\sum_{i=1}^{n}z_{i}^{2}}$$
where n is the total number of elements, $s_{0}$ is the set of all spatial weights, $z_{i\;}$ is the deviation of factor i and its average value ($x_{i}-\bar{x}$), and $w_{i,j}$ is the spatial weight between factors i and j.
(3)$$\;S_{0}=\sum_{i=1}^{n}\sum_{j=1}^{n}w_{i,j}z_{i}z_{j}$$

The *Z* score is calculated as follows:(4)$$Z_{i}=\frac{I-E\left[I\right]}{\sqrt{V\left[I\right]}},\;E\left[I\right]=-1/\left(n-1\right),V\left[I\right]=E\left[I^{2}\right]-E{\left[I\right]}^{2}$$

Before the spatial analysis, it is necessary to determine the spatial weight matrix and define the geographical distance spatial matrix *W*. The main diagonal elements of *W* are all 0, and the internal element is $w_{i,j}$. Since elements with closer distances are more likely to produce a spatial correlation effect, its expression is as follows:(5)$$w_{i,j}\;=\{\;\begin{array}{c} 1/d_{i,j},i\;\mathrm{and}\;j\;\mathrm{are}\;\mathrm{adjacent}\;\\ 0,i\;\mathrm{and}\;j\;\mathrm{are}\;\mathrm{not}\;\mathrm{adjacent} \end{array}$$
where $d_{i,j}$ represents the distance between the centroids of province i and province j.

#### 3.4.2. Regression Variables 

The environmental regulation (ER) and the square of the environmental regulation (ER^2^) are selected as explanatory variables to investigate the possible linear relationship or nonlinear relationship between environmental regulation and infrastructure investment efficiency. Through the theoretical discussion, it has been illustrated that opening up, industrial structural changes, and technological innovation have an impact on infrastructure investment efficiency. Opening up can help companies in the infrastructure sector access innovative technology and investment, while also producing a pollution shelter effect. The indicator percentage of total imports and exports to GDP is selected to represent the extent of opening up [55]. Industrial structural change is a consequence of green innovation and improvements in production efficiency, which are reflected by the indicator percentage of output value from tertiary industry to GDP [35]. Technological innovation can directly improve the production efficiency of enterprises and is represented by the number of patent applications granted per thousand people indicator [52]. In summary, the aforementioned three indicators are selected as control variables. The variable definitions are listed in Table 4.

#### 3.4.3. Spatial Econometric Models

Infrastructure investment efficiency (IIE) and numerous influencing factors have complex correlations with spatial effects; therefore, a spatial econometric model considering the spatial interactions of variables is adopted. The research model is proposed based on the research methods of Anselin [64] and Elhorst [65]. The spatial Durbin model (SDM) is a benchmark model considering both the spatial effects of dependent variables and the spatial autocorrelation of independent variables [66]:(6)$$y_{it}=\rho \sum_{j=1}^{n}W_{ij}y_{jt}+\beta X_{it}+\lambda \sum_{j=1}^{n}W_{ij}X_{jt}+\delta W_{ij}\mu_{it}+\varepsilon +c_{i}+\mu_{t}$$

In the formula, $y_{it\;}$ is the dependent variable, *X* is the influencing factor matrix, including the independent variable and control variables, $\beta_{it}$ is their influence coefficient, *W* is the spatial weight matrix in Formula (5); ρ and λ are the spatial autoregressive coefficient and the spatial error coefficient of the dependent variables; $\mu_{it\;}$ and $\varepsilon_{\begin{array}{c} it\; \end{array}}$are random error term and random error term obeying normal distribution, respectively. When ρ≠0 and λ=0, it is a Spatial Lag Model (SAR); when δ≠0 and ρ=0, it is a spatial error model (SEM); when ρ≠0 and λ≠0 and δ=0, it is a spatial Durbin model (SDM). Model selection is based on the methods of Anselin [64] and Yao, Xu, and Huang [54]. Firstly, the spatial autocorrelation test is implemented to determine whether the spatial econometric model needs to be applied. If the index exhibits spatial effects, the appropriate spatial econometric model is selected by further LM test and LR test.

Partial differential decomposition of the spatial spillover effect of the model is carried out to study the spatial spillover effect correlation between the central regions and the neighboring regions [66]; the formula is:(7)$$(i_{n}-\rho W)Y=\;\beta X_{it}+\lambda \sum_{j=1}^{n}WX+\delta W\mu +\varepsilon +c+\mu ,\;i_{n}={\left[1,0,0\ldots 0\right]}_{n}$$

Set the formula as:(8)$$(i_{n}-\rho W)=B\left(W\right),\;B\left(W\right)(i_{n}\beta_{r}-W\lambda_{r})\;S_{r}\left(W\right)$$

Then, there is:(9)$$Y\sum_{r=1}^{k}S_{r}\left(W\right)X_{r}+B\left(W\right)\;(\varepsilon +c+\mu )$$

On this basis, the spatial effects are decomposed into direct effect and indirect effect:(10)$$\frac{\partial Y}{\partial x_{1}}=\left[\begin{array}{c} S_{r}{\left(W\right)}_{11}\\ S_{r}{\left(W\right)}_{12}\\ \vdots \\ S_{r}{\left(W\right)}_{1n} \end{array}\right]=\left[\begin{array}{c} B{\left(W\right)}_{11}+\sum_{k=1}^{n}B{\left(W\right)}_{1k}W_{k1}\lambda_{x_{1}}\\ \sum_{k=1}^{n}B{\left(W\right)}_{2k}W_{k1}\lambda_{x_{1}}\\ \vdots \\ \sum_{k=1}^{n}B{\left(W\right)}_{nk}W_{k1}\lambda_{x_{1}} \end{array}\right]$$
(11)$$Direct\;Effect=B{\left(W\right)}_{11}+\sum_{k=1}^{n}B{\left(W\right)}_{1k}W_{k1}\lambda_{x_{1}}$$
(12)$$Indirect\;Effect=\sum_{k=1}^{n}B{\left(W\right)}_{2k}W_{k1}\lambda_{x_{1}}+\cdots +\sum_{k=1}^{n}B{\left(W\right)}_{nk}W_{k1}\lambda_{x_{1}}$$
where direct effect represents the influence of independent variables in the region on the dependent variable in the region, and indirect effect represents the influence of independent variables in neighboring regions on the dependent variable in the region.

The spatial lag model (SAR) is implemented to investigate the spatial effect triggered by the flow of production elements between provinces on infrastructure investment efficiency. The spatial error model (SEM) is used to study the spatial effect of infrastructure investment efficiency triggered by the disturbance of resource conditions in each province. The spatial Durbin model (SDM) is based on the two abovementioned models. The formulas for the three are as follows: 

Spatial lag model (SAR): (13)$$\;IIE_{it}=\alpha_{i}+\rho W\;IIE_{it}+\beta_{1}ER_{it}+\beta_{2}Open_{it}+\beta_{3}Ind_{it}+\beta_{4}Inn_{it}+\mu_{t\;}+\delta_{i\;}+\varepsilon_{it\;},\;\varepsilon_{it\;}\sim N\left(0,\sigma_{it}^{2}I_{n}\right)$$
(14)$$\;IIE_{it}=\alpha_{i}+\rho W\;IIE_{it}+\beta_{1}ER_{it}+\beta_{2}ER_{it}{}^{2}+\beta_{3}Open_{it}\;+\beta_{4}Ind_{it}+\beta_{5}Inn_{it}+\mu_{t\;}+\delta_{i\;}+\varepsilon_{it\;},\;\varepsilon_{it\;}\sim N\left(0,\sigma_{it}^{2}I_{n}\right)$$

Spatial error model (SEM):(15)$$\;IIE_{it}=\alpha_{i}+\beta_{1}ER_{it}+\beta_{2}Open_{it}\;+\beta_{3}Ind_{it}+\beta_{4}Inn_{it}+\mu_{t\;}+\delta_{i\;}+\varepsilon_{it\;},\varepsilon_{it\;}=\lambda W\varepsilon_{it\;}+\varphi_{it},\;\varphi_{it}\sim N\left(0,\sigma_{it}^{2}I_{n}\right)$$
(16)$$\begin{array}{c} \;IIE_{it}=\alpha_{i}+\beta_{1}ER_{it}+\beta_{2}ER_{it}{}^{2}+\beta_{3}Open_{it}\;+\beta_{4}Ind_{it}+\beta_{5}Inn_{it}+\mu_{t\;}+\delta_{i\;}+\varepsilon_{it\;},\varepsilon_{it\;}=\lambda W\varepsilon_{it\;}+\\ \varphi_{it},\;\varphi_{it}\sim N\left(0,\sigma_{it}^{2}I_{n}\right) \end{array}$$

Spatial Durbin model (SDM):(17)$$\begin{array}{c} \;IIE_{it}=\alpha_{i}+\rho W\;IIE+\beta_{1}ER_{it}+\beta_{2}Open_{it}\;+\beta_{3}Ind_{it}+\beta_{4}Inn_{it}+\theta_{1}WER_{it}+\theta_{2}WOpen_{it}\;+\theta_{3}WSec_{it}+\theta_{4}WInn_{it}\\ +\theta_{4}WInn_{it}+\mu_{t\;}+\delta_{i\;}+\varepsilon_{it\;},\;\varepsilon_{it\;}\sim N\left(0,\sigma_{it}^{2}I_{n}\right) \end{array}$$
(18)$$\begin{array}{c} \;IIE_{it}=\alpha_{i}+\rho W\;IIE+\beta_{1}ER_{it}+\beta_{2}ER_{it}{}^{2}+\beta_{3}Open_{it}\;+\beta_{4}Ind_{it}+\beta_{5}Inn_{it}+\theta_{1}WER_{it}+\theta_{2}WER_{it}{}^{2}\;+\theta_{3}WOpen_{it}\;\\ +\theta_{4}WSec_{it}+\theta_{5}WInn_{it}+\mu_{t\;}+\delta_{i\;}+\varepsilon_{it\;},\;\varepsilon_{it\;}\sim N\left(0,\sigma_{it}^{2}I_{n}\right) \end{array}$$

In the formulas, *β* and *θ* are influence coefficients, *W* is the spatial weight matrix, and *ρ* is the spatial regression coefficient. 

## 4. Results

### 4.1. Analysis of the Environmental Regulation

After the weight of each indicator is measured using the entropy weight method, the results of the environmental regulation intensity are shown in Figure 3a. From 2008 to 2020, the environmental regulation standards of various provinces and municipalities became increasingly strict. It is clear that the central and local governments had become increasingly concerned about ecological damage and made significant efforts to mitigate it. The measurement results show that the intensity of environmental regulation varies greatly among provinces. The results for Shanxi Province in 2008 were significantly higher than those of other provinces; the pollution charges and investment in environmental pollution control in Shanxi Province accounted for the highest proportion of the GDP. In recent years, the intensity of environmental regulation in Shanxi has exhibited a decreasing trend, and has gradually returned to the national average level, indicating that the environment of Shanxi Province has been gradually improving since 2008. 

In ArcGIS, the natural breaks classification method is used to observe the results in Figure 3b. In general, the geographical distribution of environmental regulation intensity is uneven, which is higher in the east than the west, and higher in the north than the south. Most provinces with relatively high environmental regulation intensities are clustered in the east and north; only Xinjiang, with relatively high environmental regulation intensity, is located in the west and south.

After calculation, from 2008 to 2020, the global Moran’s *I* of environmental regulation in each year was greater than 0 and significant, indicating that environmental regulation in China was not randomly distributed, but exhibited a relatively distinct spatial correlation. Provinces with high environmental regulation intensity are surrounded by provinces with high environmental regulation intensity, and provinces with low environmental regulation intensity also exhibit agglomeration characteristics.

Local spatial correlation analysis is carried out to ascertain whether each province belongs to a relatively high-intensity or low-intensity environmental regulation region. The Moran’s *I* scatter plot can reflect the correlation characteristics of adjacent units in local areas, and contains four quadrants. The objects in the first and third quadrants (the HH or LL regions) exhibit positive spatial correlations, while the objects in the second and fourth quadrants (the LH and HL regions) exhibit negative spatial correlations, which correspond to spatial aggregation and spatial dispersion, respectively. The Moran’s *I* scatter plots for 2009, 2013, 2017, and 2020 (see Figure 4) are selected, and it is observed that characteristics of spatial agglomeration are significant. 

### 4.2. Analysis of the Infrastructure Investment Efficiency

In Figure 5a, it can be seen that the infrastructure investment efficiency in all provinces and municipalities fluctuates greatly, exhibiting an overall growth trend, indicating that—with increases in China’s infrastructure investment scale—investment control is becoming more refined and effective. 

As is shown in Figure 5b, the natural breaks classification method is implemented in ArcGIS to assess the results. In general, infrastructure investment efficiency is relatively evenly distributed, with strong spatial aggregation characteristics. Besides, the infrastructure investment efficiency in China is generally high in the east and low in the west, but Shanxi is an exception, which is located in the north and has relatively higher infrastructure investment efficiency. 

In Figure 6, the objects in the first and third quadrants (the HH or LL regions) exhibit spatial aggregation characteristics, while the objects in the second and fourth quadrants (the LH and HL regions) exhibit spatial dispersion characteristics. It can be seen that more than 60% of provinces and municipalities are distributed in the first quadrant (HH) and the third quadrant (LL), indicating that the infrastructure investment efficiency exhibits spatial aggregation. The spatial-temporal transition method adopted by Rey [67] is further implemented to describe the spatial-temporal evolution of Moran’s scatter plots in Figure 6. The years 2009, 2017, and 2020 are selected for analyzing temporal and spatial transitions, as shown in Table 5. Only 6 out of 30 provinces exhibit the space–time transition phenomenon, indicating that the infrastructure investment efficiency of the vast majority of provinces and municipalities, along with their neighbors, has a high degree of spatial stability. To sum up, the infrastructure investment efficiency exhibits stable spatial agglomeration characteristics, which is suitable for the spatial econometric model.

### 4.3. Analysis of the Spatial Regression 

The formula that contains only ER is defined as a linear model, and the other formula—containing ER and ER^2^—is defined as a nonlinear model. The LM test results indicate that both models are significant and have spatial effects, as shown in Table 6. The LR test results show that both linear and nonlinear models reject the degradation of SDM models into SAR and SEM models (*p* = 0.000). The Hausman test results show that both models are suitable for the fixed effect model (*p* = 0.000). After implementing the LR test to choose the temporal period fixed, spatial fixed, and temporal period–spatial fixed models, the temporal period–spatial fixed model is found to be the most applicable; therefore, the temporal period–spatial fixed SDM model is implemented in this study. 

#### 4.3.1. Regression Results of the Linear Model

Under the spatial Durbin model, the intensity of environmental regulation (ER) is significant at a 1% confidence level, and the correlation coefficient is positive, indicating that environmental regulation plays a key role in promoting infrastructure investment efficiency (as shown in Table 7).

The ρ-value of the SDM model is significantly positive at the level of 5%, indicating that environmental regulation promotes the spatial spillover of infrastructure investment efficiency. The spatial effects in the SDM model are divided into direct effects and indirect effects in Table 7. Direct effect refers to the spatial effect of variable changes on the dependent variable of the region itself, while indirect effect refers to the spatial effect of variable changes on the dependent variable of the surrounding regions. The total effects are the combination of direct effect and indirect effect. The direct effect and total effects of environmental regulation are significant, indicating that environmental regulation mainly produces spatial effects through the infrastructure investment efficiency of the region itself rather than that of surrounding regions. The total effects of ER are 0.0362, suggesting that a 1% increase in environmental regulation will promote the focal provinces’ infrastructure investment efficiency by 0.0362%. Direct effect, indirect effect, and the total effects of open are significant at the 1% confidence level, indicating that foreign investment promotes infrastructure investment efficiency through capital spillover in the region and surrounding areas.

#### 4.3.2. Regression Results of the Nonlinear Model

From Table 8, it can be seen that the coefficient of ER is significantly positive and that of ER^2^ is significantly negative, indicating that the impact of environmental regulation on infrastructure investment efficiency presents an inverted U-shaped relationship. In other words, environmental regulation has a threshold effect on infrastructure investment efficiency. 

An appropriate intensity of environmental regulation is beneficial to improving the infrastructure investment efficiency, but too strong an intensity of environmental regulation will inhibit infrastructure investment efficiency. Furthermore, the Durbin term of ER presents a U-shaped relationship, indicating that the improvement of ER in neighboring provinces first inhibits and then promotes the infrastructure investment efficiency in the focal province. Therefore, the impact of environmental regulation on the focal province and adjacent provinces is the opposite, which verifies the existence of a signal transmission mechanism. The improvement of environmental regulation will reduce pollution spillover, leading to the weakening of pollution signals in neighboring areas; neighboring provinces will then adopt moderately flexible environmental policies. The direct effects, indirect effects, and total effects of environmental regulation in the SDM model are further studied (see Table 8). The direct and indirect effects of ER are opposite, indicating that the impact of environmental regulation on infrastructure investment efficiency in a province is opposite to that of its neighboring provinces, which verifies that there is a substitution effect of environmental regulation on the spatial spillover of infrastructure investment efficiency. Neighboring provinces with looser environmental regulations prefer to introduce green technologies rather than develop them themselves, which uses up funding for innovation.

## 5. Discussion

China has invested a huge amount in infrastructure and carried out large-scale urban construction in recent years, which has put significant pressure on the eco-environment. Large-scale infrastructure investment needs to be used efficiently, and environmental regulation plays an indispensable role in this process. EKC theory holds that the relationship between the environment and infrastructure quality presents an inverted U-shaped relationship, and the Porter hypothesis posits that appropriate environmental regulation will promote production efficiency. This study aims to explore the spatial relationship between environmental regulation and infrastructure investment efficiency, and the results are expected to provide useful suggestions for ecological protection and the efficient utilization of infrastructure investment.

We mainly interpreted the discussion from four aspects: (1) measurement results of environmental regulation and its spatial distribution; (2) measurement results of infrastructure investment efficiency and its spatial distribution; (3) regression results of the linear model and nonlinear model; and (4) spatial effects of the models.

Firstly, we analyze the results of environmental regulation and compare them with other literature. On the whole, the intensity of environmental regulation in various provinces and municipalities in China keeps rising, but the spatial distribution is uneven and has an aggregation effect. The result supports Hypothesis 3 and is consistent with the findings of Feng et al. [68] and Wang, Xia, and Xia [23]. We compared the spatial distribution with other literature: (1) Spatial pattern. The spatial pattern of environmental regulation in China is “higher east than west and higher north than south”, the results of Zhang, Liu, and Zhang [63], Fu et al. [69], and Hu and Wang [70] support this phenomenon. (2) The difference with other literature. At the same time, it has been shown that the eastern region and the northern region possess more pollution emissions [61]. Theoretically, a large amount of pollution emission is due to loose environmental regulation, which seems to be inconsistent with the research conclusion. (3) The reason for the difference. Given the large number of heavily polluting industries and resource enterprises in China’s northern provinces, strong environmental controls are necessary; for example, in Shanxi Province, pollution discharge fees and pollution control investment accounted for a high proportion of the GDP. Besides, the eastern region, on the other hand, with a large number of industrial polluting enterprises, possesses strong green innovation ability and strong environmental control [71]. To sum up, the eastern and northern regions present a coexistence pattern of high pollution and strong environmental governance. Some literature supports this view, China’s high-polluting enterprises are mainly distributed in the east, and strong environmental regulations will inhibit pollution transfer [69]; it is found that the pollution in the northern region is stronger, but the stronger environmental regulation has also achieved certain effects, and environmental regulation plays a more significant role in the eastern and northern cities [62]. (4) A special case. Xinjiang, located in the northwest, possesses higher environmental regulation intensity, with higher pollution discharge fees and pollution control investment/GDP. The reason for this may be the existence of large-scale resource mining in Xinjiang, where the government has invested heavily in controlling opencast coal mining and land pollution [72]. 

Secondly, we analyze the results of infrastructure investment efficiency and compare them with other literature. The infrastructure investment efficiency of all provinces and municipalities in China presents an increasing trend with spatial aggregation features on the whole. The result supports Hypothesis 3 and matches the findings of Cheng and Lu [4] and Chen, Shen, Zhang, Li, and Ren [5]. This result indicates that China’s investment control in the field of infrastructure construction is gradually being refined, and technological innovation is playing an increasingly important role [71]. The provinces and municipalities with higher infrastructure investment efficiency are mainly clustered in the eastern areas, which is consistent with the results of Cheng and Lu [4]. There are three possible reasons for this phenomenon: (1) The convenience of trade between eastern areas and other countries is conducive to obtaining green investment in the infrastructure industry and to the introduction of green technology; (2) the eastern regions possess higher ecological efficiency and a more environmentally friendly infrastructure industry [54,73], which are also conducive to infrastructure investment efficiency; and (3) the eastern regions, with higher energy efficiency, possess the advantage of producing more benefits per unit of energy consumption, which is conducive to saving the costs of infrastructure construction and operation. In other literature, although the infrastructure investment efficiency based on city-level data also presents the higher feature of eastern coastal areas, the distribution in the whole country is more dispersed [5,12]. This phenomenon further verifies the advantages of the eastern coastal areas in infrastructure investment efficiency, while the overall infrastructure investment efficiency in the central and western regions needs to be improved. We also noticed the higher infrastructure investment efficiency in Shanxi Province, located in the north, which indicates that the resource management measures and green technology innovation adopted in Shanxi have played a positive role, for example, promoting green mining infrastructure [74]. Both environmental regulation and infrastructure investment efficiency present spatial aggregation characteristics; therefore, the spatial econometric model is implemented to conduct empirical research. According to the theoretical discussion, the impact of environmental regulation could be a linear model or a nonlinear model, and the analysis will be carried out next. 

Thirdly, we analyze the regression results of the linear model and nonlinear model and compare them with other literature. The empirical results of the linear model show that environmental regulation promotes the growth of infrastructure investment efficiency, which is consistent with the findings of Yasmeen, Tan, Zameer, Tan, and Nawaz [52] and De Santis, Esposito, and Lasinio [26]. The result supports Hypothesis 1, mainly for three reasons: (1) Appropriate environmental regulations will directly guide enterprises to reduce energy consumption and improve efficiency [9]; (2) green economy subsidies given by the government to regulated enterprises help improve the efficiency [41]; (3) moderate environmental regulation encourages green technology innovation and attracts foreign investment [30,43], which are conducive to the efficiency. This result further suggests that, although environmental regulation also has certain compliance costs, these costs are offset by direct regulatory benefits and indirect benefits such as green subsidies and green technology innovation. This also explains to some extent the reason why the eastern coastal regions possess both higher environmental regulation intensity and higher infrastructure investment efficiency. In the nonlinear model, an inverted U shape is presented, which is similar to the results of Song et al. [75], Zhang, Ouyang, Ballesteros-Perez, Li, Philbin, Li, and Skitmore [30], and Wang et al. [76]. The result supports Hypothesis 2, mainly for three reasons: (1) Strict environmental regulation exerts pressure on enterprise compliance costs such as the payment of cleaner production equipment, which is not conducive to efficiency [42]; (2) strict environmental regulation will form investment barriers that are unfavorable to efficiency; (3) strict environmental regulations will crowd out enterprises’ investment in technological innovation, which is not conducive to the efficiency [41,77]. The result further suggests that large-scale infrastructure investment with appropriate environmental regulation directly saves a lot of costs, which can offset compliance costs, but too strict environmental regulation will greatly increase the operating costs of enterprises and reduce infrastructure investment efficiency. The results of Li, Tang, Tenkorang, and Shi [32] about the relationship between environmental regulation and efficiency from the perspective of corporate finance support this view.

Fourthly, we analyze the spatial effects of the models and compare the results with other literature. In the nonlinear model, the Durbin term of environmental regulation presents a U-shaped relationship. The result supports Hypothesis 4 and is consistent with the findings of Li and Du [22]. Low-intensity environmental regulation can promote infrastructure investment efficiency and restrain spatial spillover, while high-intensity environmental regulation has the opposite effect. This phenomenon is due to the large scale of infrastructure investment in China. Although the effect of flexible environmental policies is relatively prominent, it cannot stimulate green technology innovation, and it is difficult to achieve a technology spillover effect. Besides, the results of the spatial effects decomposition show that the environmental regulation of a certain province has opposing effects on infrastructure investment efficiency in its own area and neighboring areas, which also verifies the substitution effects of environmental regulation on the spatial spillover of the infrastructure investment efficiency. Surrounding provinces with lower environmental regulation will prioritize the introduction of innovative technologies from focal province regulation rather than implementing original innovations.

Among all control variables, opening up can promote infrastructure investment efficiency, which is similar to the findings of Konara et al. [78]. Opening up can help enterprises obtain foreign investment and advanced technology, promoting high-quality production. It seems that the effect of industrial structure upgrading on infrastructure investment efficiency is not very significant, which is not consistent with the findings of Zhou et al. [79] and Wang, Song, Duan, and Wang [53]. The reason for this could be that the overall utilization efficiency of China’s infrastructure investment is low [5], and the economic benefits of industrial changes can hardly offset the large amount of investment wastage. In addition, the transformation of achievements in different stages of the construction industry is not smooth [30], which will also lead to poor industrial upgrades. Technological innovation can promote the improvement of infrastructure investment efficiency, which is in line with the findings of Zhai and An [80] and Wang et al. [81], indicating that technological innovation can not only promote the improvement of infrastructure investment efficiency in focal regions but also improve the infrastructure investment efficiency through the technology spillover effect.

## 6. Conclusions

As the conflict between economic development and the carrying capacity of the environment becomes increasingly intense, effectively controlling environmental pollution while promoting infrastructure investment efficiency has become an urgent problem to solve. In this context, this research explores the influence mechanism and spatial effects of environmental regulation on infrastructure investment efficiency, and the main conclusions are as follows:

Firstly, theoretically, environmental regulation has a signal transmission mechanism and has a spatial effect on infrastructure investment efficiency through substitution effects or compensation effects. The spatial spillover of knowledge and technology is obvious and plays a crucial role in economic growth. As a result, technological innovation should be encouraged to improve efficiency and stimulate the spillover of infrastructure investment.

Secondly, from 2008 to 2020, both the environmental regulation and infrastructure investment efficiency in China increased, in general, and exhibited spatial aggregation. The eastern and northern regions in China are characterized by strong environmental regulation and high pollution, and the infrastructure investment efficiency is higher in the eastern areas. Therefore, the government should formulate environmental policies in accordance with local resource availability to achieve a “win-win” scenario for economic development and environmental protection.

Thirdly, the empirical results based on the linear model show that improvements in the environmental regulation intensity will promote infrastructure investment efficiency. The empirical results based on the nonlinear model show that, with increases in the environmental regulation intensity, infrastructure investment efficiency will increase first and then decrease. As a result, the government needs to select appropriate policies according to the inflection point to improve investment efficiency and promote technological innovation. Government departments need to implement efficient infrastructure investments and ensure that they can meet the needs of the people.

Fourthly, environmental regulation has an opposing spatial effect on the infrastructure investment efficiency in focal provinces and surrounding provinces, and the spillover effects on the infrastructure investment efficiency present a U-shaped relationship. At present, in the context of green development and high-quality economic development, the negative impact of environmental regulation on infrastructure investment efficiency should be eliminated to the highest possible extent. In addition, it is necessary to stimulate the positive impact of environmental regulation on infrastructure investment efficiency, expand the technology spillover effect, and make environmental regulation a favorable tool to improve infrastructure investment efficiency.

## Figures and Tables

**Figure 1 ijerph-20-02960-f001:**
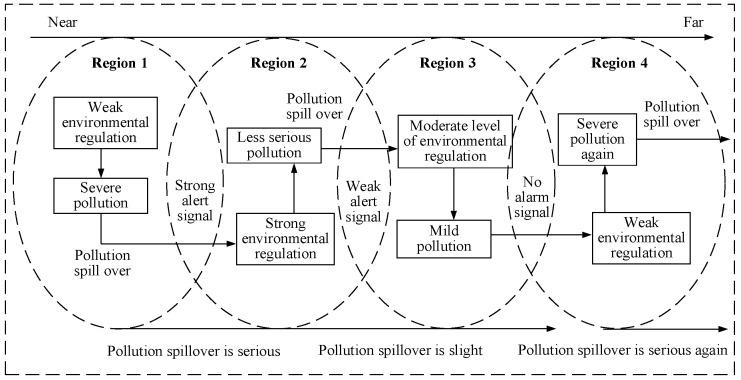
Signaling mechanism.

**Figure 2 ijerph-20-02960-f002:**
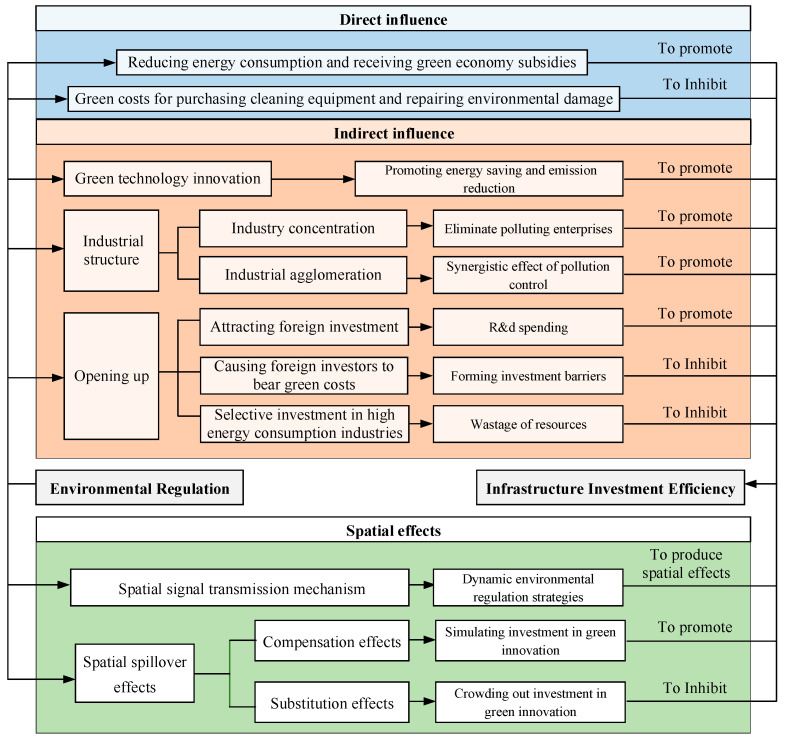
Influence mechanism of environmental regulation on infrastructure investment efficiency.

**Figure 3 ijerph-20-02960-f003:**
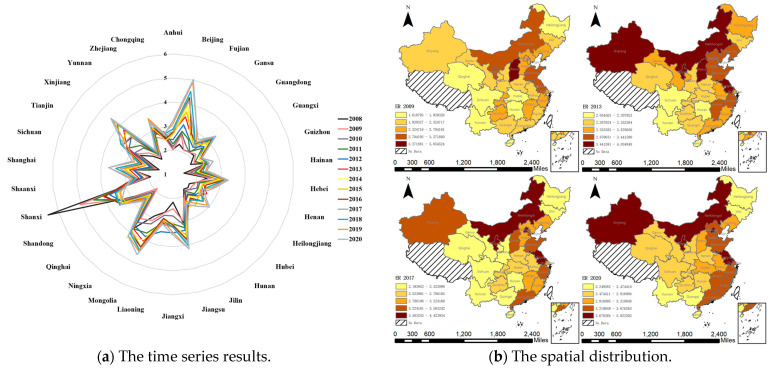
The results and spatial distribution of environmental regulation.

**Figure 4 ijerph-20-02960-f004:**
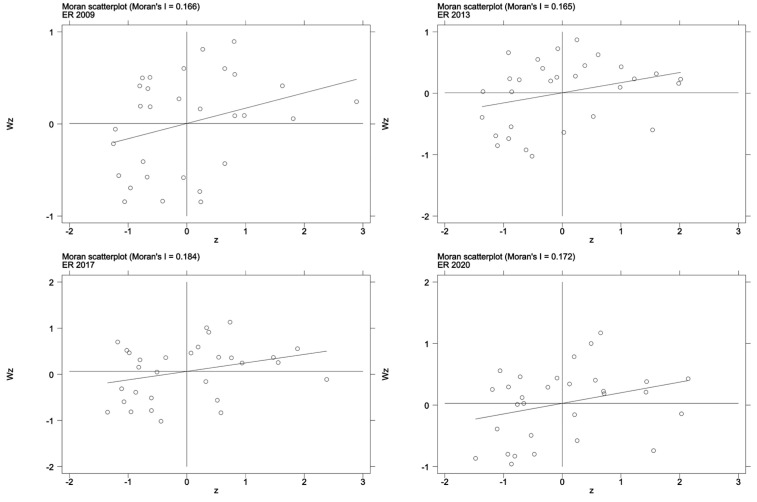
Moran’s scatter plot for environmental regulation.

**Figure 5 ijerph-20-02960-f005:**
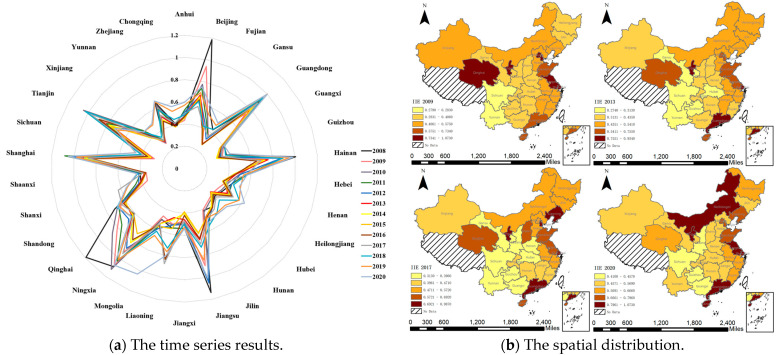
The results and spatial distribution of infrastructure investment efficiency.

**Figure 6 ijerph-20-02960-f006:**
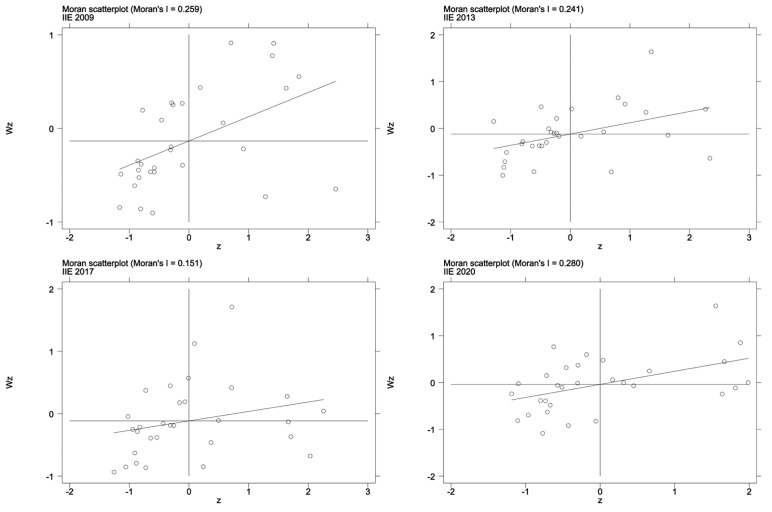
Moran’s scatter plots for the infrastructure investment efficiency.

**Table 1 ijerph-20-02960-t001:** Scope of infrastructure investment and Infrastructure evaluation index system.

Scope of Infrastructure Investment	Infrastructure Evaluation Index System	
	System	Evaluation Index
Production and supply of electricity, gas, and water; Transportation, warehousing, and postal services; Water conservancy, environment, and public facility management; Health, social security, and social welfare; Education, culture, sports, and entertainment; Information transmission, computer services, and software; Scientific research, technical services, and geological surveys; The construction industry.	Energy power system	Gas penetration rate (%)
		Electricity consumption (billion KWH)
	Water resources system	Water resources per capita (m^3^/person)
		Water penetration rate (%)
		Water supply production capacity (10,000 m^3^/day)
	Water supply and drainage system	Daily sewage treatment capacity (10,000 m^3^)
		Length of water supply pipeline (km)
		Length of drainage pipe (km)
	Road traffic system	City road lighting (1000)
		Actual road length (km)
		Bus and tram operating line network length (km)
		Number of buses per 10,000 people
	Post and telecommunications system	Number of post offices
		Urban telephone subscribers (million households)
		Internet broadband access port (1000)
	Ecological system	Green coverage rate of built-up area (%)
		Per capita green area of park (m^2^)
	Social service system	Number of libraries (100) Number of hospitals (100) Number of secondary schools (100)

**Table 2 ijerph-20-02960-t002:** Indicators for measuring infrastructure investment efficiency.

Input Indicators	Output Indicators
The fixed asset investment corresponding to each scope of industry in Table 1; The number of employees corresponding to each scope of industry in Table 1	Total GDP; Average disposable income; Urban infrastructure composite index

**Table 3 ijerph-20-02960-t003:** Environmental regulation intensity measurement index system.

First-Level Indicators	Second-Level Indicators	Basis
Government administration	Enterprises pollution charges/GDP	[58]
Regulation effects	Centralized treatment rate of sewage	[57]
	Harmless treatment rate of garbage	
	Sulfur dioxide emissions/industrial added value	
	Solid waste production/industrial added value	[20,57]
	Sewage discharge/industrial added value	
Economic input	Investment in environmental pollution control/ GDP	[51]
	Per capita GDP	[2,51,59,60]
Residents’ green life quality	Per capita park green area	[50,52]
	Coverage rate of sanitary latrine	

**Table 4 ijerph-20-02960-t004:** Variable definitions.

Variable Type	Variable Name	Symbol	Calculation
Dependent variable	Infrastructure Investment efficiency	IIE	Results from the Super-SBM model
Core variables	Environmental regulation	ER	Results from the entropy weight method
		ER^2^	
Control variables	Opening up	Open	Percentage of total imports and exports to GDP
	Industrial structure upgrading	Ind	Percentage of output value from tertiary industry to GDP
	Technological innovation	Inn	Number of patent applications granted per thousand people

**Table 5 ijerph-20-02960-t005:** Moran’s scatter plots changes of infrastructure investment efficiency.

	Pattern	2009	2017	2020
First quartile	HH	Hainan, Shanghai, Zhejiang, Shandong, **Beijing**, Tianjin, Jiangsu (7)	Hainan, Shanghai, Shandong, Zhejiang, Tianjin, Jiangsu (6)	Hainan, Shanghai, Zhejiang, Shandong, Tianjin, Jiangsu, **Liaoning**, **Ningxia** (8)
Second quartile	LH	Anhui, Gansu, Hebei, Fujian, Jilin (5)	Gansu, Anhui, Hebei, Fujian, **Inner Mongolia**, Jilin (6)	Hebei, Anhui, Gansu, Fujian, Jilin, **Inner Mongolia, Chongqing** (7)
Third quartile	LL	**Inner Mongolia**, Hunan, Guangxi, Sichuan, Shaanxi, Henan, Hubei, Guizhou, Yunnan, **Shanxi**, Heilongjiang, Xinjiang, Jiangxi, **Liaoning, Chongqing** (15)	**Beijing,** Hubei, Hunan, Guangxi, Sichuan, Shaanxi, Henan, Guizhou, Yunnan, Heilongjiang, **Liaoning**, Xinjiang, Jiangxi (13)	Shaanxi, Hunan, Guangxi, Sichuan, Henan, Hubei, Guizhou, Yunnan, **Shanxi**, Xinjiang, Heilongjiang, Jiangxi (12)
Forth quartile	HL	Qinghai, **Ningxia**, Guangdong (3)	Qinghai, **Ningxia**, Guangdong, **Shanxi, Chongqing** (5)	**Beijing**, Guangdong Qinghai (3)

**Table 6 ijerph-20-02960-t006:** Test results.

Test Statistics	Linear Model		Nonlinear Model	
	Statistic	*p*-Value	Statistic	*p*-Value
LMLag	26.570	0.000	27.196	0.000
Robust LMLag	7.517	0.006	6.988	0.008
LMErr	64.171	0.000	64.240	0.000
Robust LMErr	45.118	0.000	44.032	0.000
lrtest SDM SAR	95.94	0.000	95.75	0.000
lrtest SDM SAR	98.73	0.000	98.96	0.000
Hausman test	65.75	0.000	64.14	0.000

**Table 7 ijerph-20-02960-t007:** Linear model test results.

Variable	SDM	Direct Effects	Indirect Effects	Total Effects
ER	0.0784 *** (5.42)	0.0808 *** (5.43)	−0.0445 * (−1.89)	0.0362 * (1.86)
Open	0.3418 *** (9.54)	0.3358 *** (9.61)	0.1106 * (1.77)	0.4464 *** (7.09)
Ind	0.1547 (1.10)	0.1753 (1.28)	−0.1101 (−0.44)	0.0652 (0.26)
Inn	0.0019 * (1.70)	0.0018 * (1.73)	0.0026 (1.02)	0.0045 * (1.77)
W*ER	−0.0378 (−1.12)			
W*Open	0.1729 ** (2.22)			
W*Ind	−0.0908 (−0.35)			
W*Inn	−0.0031 (−1.12)			
ρ	0.1642 ** (2.14)			
Rbar^2^	0.5177			
log.likelihood	174.6998			

Note: *** *p* < 0.01, ** *p* < 0.05, * *p* < 0.10.

**Table 8 ijerph-20-02960-t008:** Nonlinear model test results.

Variable	SDM	Direct Effect	Indirect Effect	Total Effects
ER	0.2761 *** (3.45)	0.2801 *** (3.48)	−0.0408 * (−1.81)	0.2393 (1.21)
ER^2^	−0.0302 ** (−2.52)	−0.0305 ** (−2.53)	−0.0033 (−0.12)	−0.0339 (−1.06)
Open	0.3054 *** (7.98)	0.3035 *** (8.02)	0.1042 (1.47)	0.4078 *** (5.50)
Ind	0.2449 (1.67)	0.2444 * (1.81)	−0.1296 (−0.52)	0.1148 (0.45)
Inn	0.0019 * (1.78)	0.0019 * (1.80)	0.0020 (0.80)	0.0040 (1.55)
W*ER	−0.4125 * (−1.86)			
W*ER^2^	0.0711 ** (2.22)			
W*Open	0.1768 ** (2.01)			
W*Ind	−0.1369 (−0.49)			
W*Inn	−0.0025 (−0.90)			
ρ	0.1736 ** (2.27)			
Rbar^2^	0.5191			
log.likelihood	177.9458			

Note: *** *p* < 0.01, ** *p* < 0.05, * *p* < 0.10.

## Data Availability

The data that support the findings of this study are openly available via the National Bureau of Statistics of PRC at https://data.stats.gov.cn/easyquery.htm?cn=C01 (accessed on 1 June 2022), and the EPS DATA at https://www.epsnet.com.cn/index.html#/Home (accessed on 1 June 2022).

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
