# Peer review of "Does Environmental Regulation Promote the Infrastructure Investment Efficiency? Analysis Based on the Spatial Effects"

_ijerph, 2023, doi:10.3390/ijerph20042960_

Round 1

Reviewer 1 Report

This paper analyses the relationship between environmental regulation and the infrastructure investment efficiency from the spatial perspective, and the workload of this article is very large. However, this paper still needs to be improved in some aspects. My detailed comments are as follows:

(1)   The data in the article is from 2008 to 2017, which is older than other papers? Why not choose newer data?

(2)   Line 240 of the text mentions that infrastructure includes education, culture, sports, health and other social service facilities, etc. However, it is not reflected in the construction indicators of the infrastructure evaluation index system, why? What is the basis for the establishment of the index system?

(3)   The relationship between the environment and infrastructure investment is mentioned several times in the introduction of this paper and in other literatures, but environmental indicators are not reflected in this paper when measuring the efficiency of infrastructure investment in the Table 2, why?

(4)   The index system for environmental regulation in line 295 is not sufficiently visual as it is expressed in words only, could it be described using tables?

(5)   When describing the spatial characteristics of the intensity of environmental regulation in Figure 3(b) in line 396, it is mentioned that there is a spatial pattern of "higher east than west and higher north than south", what are the results obtained from existing studies? Are there any differences with existing studies? What are the reasons for this? Also, the text states that Chongqing, Jiangxi, Xinjiang and Qinghai provinces do not conform to the spatial pattern described above, what are the reasons for this? It is suggested that this be added to the discussion section.

(6)   The description of the spatial characteristics of the efficiency of infrastructure investment in Figure 5(b) in line 425 is poor and contains little valuable information, and it is suggested that relevant content be added to the discussion section.

Reviewer 2 Report

General comments

This research focuses on whether environmental regulation promotes the efficiency of infrastructure investment. The content of this topic does not highlight the necessary value of the research sufficiently. The introduction, methods, results and conclusions need to be improved to enhance the practical significance. In addition, some sentences need to be reexamined, the exact information of the research cannot be conveyed, and revision is needed.

 Abstract

The abstract is too succinct and fails to reflect the entire study. The author should follow the IMRAD style to write a nice abstract. At the same time, pay attention to the words and sentences of the abstract, not only to show the research results, but also to highlight the value and significance of the article. 

Introduction 

- The introduction structure is good, the literature review is sufficient, and the contribution of the research can be clarified, but the significance of the research is not enough.

- There is no coherence in the description.

- The author need to avoid irrelevant discussions.

- There is no clear research hypothesis.

- This part does not explain clearly the sentence the construction and operation of infrastructure do not match the carrying capacity of resources and ecology in China, which produces significant pressure on the environment in the abstract.

Materials and Methods

- This section is poor in terms of information and presentation.

- The process of several methods is unclear. It is suggested that the author add a subsection to describe the method correctly.

Results and discussion

-This section is poor and haphazard.

-There is no planning for presenting the findings properly.

-The description lacks coherence due to procedural weakness.

Round 2

Reviewer 2 Report

Thank you for the revision and response!

Author Response

Thanks for your kind response!